# Cytogenetic Effects in Patients after Computed Tomography Examination

**DOI:** 10.3390/life12121983

**Published:** 2022-11-27

**Authors:** Elizaveta Neronova, Sergei Aleksanin

**Affiliations:** Nikiforov Russian Center for Emergency and Radiation Medicine (NRCERM), 194044 Saint-Petersburg, Russia

**Keywords:** non-cancer patients, ionizing radiation, CT examination, chromosome aberration, dicentric, genotoxic effects

## Abstract

Millions of people around the world are exposed to low doses of ionizing radiation from diagnostic computed tomography (CT) scans. Currently available data on the potential cancer risk after CT scans are contradictory and therefore demand further investigations. The aim of the current study was to obtain estimations of genome damage after CT scans in 42 non-cancer patients and to conduct a comparison of the results with 22 control subjects. The frequency of dicentric ring chromosomes and chromosome breaks was significantly increased in irradiated patients compared to the controls. The distribution of dicentrics among the cells demonstrated non-Poisson distribution that reflected non-uniform and partial-body radiation exposure. A fraction of patients followed Poisson distribution, which is typical for uniform whole-body exposures. Some patients demonstrated a level of dicentrics similar to the control subjects. The individual variations in the frequency and dicentric distribution suggested complex mechanisms of chromosome aberration induction and elimination that could be associated with individual radiosensitivity, as well as previous diagnostics that used ionizing radiation or the redistribution of small fractions of irradiated lymphocytes within the circulatory pull. In conclusion, CT scans may cause genome damage and possible increases in cancer risk. The introduction of a specific follow-up of such patients, especially in the case of repeated CT scans, is suggested.

## 1. Introduction

The human population is constantly exposed to different types of radiation (from both natural and manmade sources). It is estimated that 80.1% of the total radiation dose subjected to humans is caused by natural radiation sources, while the doses of background radiation vary in different regions worldwide, from 1.25 mSv to 4.5 mSv annually [1,2].

Currently, a significant part of the population is exposed to ionizing radiation as the result of medical diagnostic and therapeutic procedures, such as cancer patients and patients with somatic diseases. In medical practice, subjects are exposed to ionizing radiation in a wide age range: from neonates to elderly people. Computed tomography (CT) scans started to become a widespread diagnostic tool during the 1980s, and their use from then on increased dramatically. In Russia, for example, a rapid increase in the number of these diagnostic imaging procedures occurred, from 1.3 million in 2003 to 11.7 million in 2018. As a result, the contribution of CT scans to the dose of incurred medical radiation increased from 5.7% to 54.0% [2]. Similar trends have been observed in other countries. For example, a report shows that 67 million CT scans were performed in 2006 compared to the 84 million performed in 2016 in the USA. Further, this made up 63% of the collective dose from medical imaging procedures in 2016, compared to 50% in 2006 [3]. In Europe, there was an increase in the number of patients who underwent CT scans by about 41% between 2015 and 2018 [4]. Doses from different types of CT examinations varied from 0.6 mSv to 31 mSv, which gave patients the equivalent of a background radiation dose of (depending on the region) 2 months to 10 years [5]. Due to the steady growth in the use of CT scans worldwide, there is also growing concern about the increased risk of cancer as a result of diagnostic radiation. The risk of oncopathology in children after CT scans is mostly the focus of research due to an increased lifetime radiation risk relative to those for adults [6,7,8]. Zhang et al. [9] conducted a population-based case–control study among adults. They provide the first direct evidence that CT scanning and nuclear medicine examinations are associated with an increased risk of thyroid cancer. Shao et al. [10] conducted a nested case–control study in an adult cohort and reported that CT studies may be associated with the risk of thyroid cancer and leukemia; further, in patients younger than 45 years, an increased risk of the development of non-Hodgkin’s lymphoma was also found. However, Lee et al. [11] suggested that the risk of radiation-induced thyroid cancer following CT scanning in adults may have been overestimated in observational studies, due to medical surveillance-related biases. The high incidence of thyroid cancer in adults exposed to ionizing radiation during CT scanning can be largely explained by the confounding effect of the healthcare utilization rate [11]. Therefore, the results of cancer risk assessments in adult patients are quite contradictory. However, taking into account the fact that millions of people are exposed to medical diagnostic radiation, and considering that the Biologic Effects of Ionizing Radiation VII (BEIR) model postulates [12] that there is no safe level of ionizing radiation exposure, then it must be noted that even the smallest exposure carries some level of cancer risk. As such, the effects of CT examination on humans need to be examined further.

Ionizing radiation induces various types of DNA damage that lead to alterations of genetic information, such as chromosome aberrations. The analysis of unstable chromosome aberrations in peripheral blood lymphocytes is one of the most widely used biomarkers of DNA damage in humans for investigating the genotoxic effects of environmental, lifestyle, and occupational factors. It is well known that unstable chromosome aberrations, namely dicentric and ring chromosomes, in human lymphocytes have been used for several decades as a biological indicator of radiation exposure, as well as a biological instrument for dose estimation [13]. Other types of chromosome aberrations can serve as predictors of carcinogenic health effects. Thus, the statistical association between chromosome aberration frequency in peripheral blood lymphocytes and cancer risks detected in different groups of people supports the hypothesis that chromosome aberration is a predictor of cancer [14]. These facts suggest the usefulness of including chromosome aberrations as biomarkers of genotoxic effects in persons exposed to medical ionizing radiation. 

The aim of this investigation was, therefore, to evaluate unstable chromosome aberrations in peripheral blood lymphocytes in patients following CT examination.

## 2. Materials and Methods

### 2.1. Participants

Forty two non-cancer patients (15 women and 27 men), aged between 31 and 72 years, undergoing CT scanning were enrolled in this study. The control group consisted of 22 non-cancer patients (5 women and 17 men) of the same age and health status, naïve to CT scanning. All participants were examined and treated due to diseases of the digestive, respiratory, or cardiovascular systems at the Nikiforov Russian Center for Emergency and Radiation Medicine (Saint-Petersburg, Russia). Patients were interviewed before the study. The inclusion criteria were as follows: patients over 18 years of age; patients who had no contact with genotoxic factors. Exclusion criteria: patients with a history of cancer, leukemia, or lymphoma; patients who underwent interventional radiology or nuclear medicine procedures. 

### 2.2. CT Scans

CT scans of the brain (3 patients), abdomen (11 patients) or chest (28 patients) were performed with (17 cases) or without (25 cases) contrast. This was achieved using a SOMATOM EMOTION ECO CT scanner (16-slice configuration) with a tube voltage of 100–130 kV, mAs = 70–120, and pitch = 0.8–1.5 mm. The effective radiation dose was calculated by the computational dosimetry system EDEREX (effective dose estimation at Roentgen examinations). Further, it was set according to methodical guidance [15] using data regarding age, gender, initiation, and the end position of the CT scan. The dose length product (DLP) was calculated according to the CT dose index (CTDI), which was set uniformly in the CT scanner with the length (L) of the axial CT scanning range of a body, as below:DLP = CTDI (mGy) × L (cm).

For the chest, doses ranged between 1.5 and 3 mSv (with a contrast of 4–10 mSv); for the abdomen, they ranged between 2.5 and 5 mSv (with contrast of 4.4–15 mSv); and for the brain, they ranged between 2 and 3 mSv.

The patients’ characteristics and effective doses are shown in Table 1.

There was no significant difference between the levels of patient exposure in Russia and other countries [16].

### 2.3. Peripheral Blood Culture for Unstable Chromosome Aberration Analysis

Peripheral blood samples were collected by venipuncture in a sodium heparin vacutainer. In exposed patients, blood was sampled within 3–14 days after the CT scan in order to avoid any bias caused by the late DNA repair process of the patients [17]. Two whole blood samples (0.5 mL) per person were cultured for 54 h at 37 °C in centrifuge tubes containing medium RPMI 1640 (Biolot, Saint-Petersburg, Russia). These were supplemented with 20% fetal calf serum (Biolot), 50 μg phytohemagglutinin (PANEKO, Moscow, Russia), and 20 mM of glutamine. After 46 h of cultivation, colchicine was added at a final concentration of 0.05 μg/mL. Metaphase cells were harvested with a hypotonic 0.075 M potassium chloride solution, followed by three fixation steps (3:1 ethanol–acetic acid). Cell suspensions were spread onto slides. Slides were dried and stained with a Giemsa 5%. Cells were observed under a bright-light microscope at 1000× magnification. For each patient, up to 1200 metaphases were analyzed. Four categories of chromosome aberrations were evaluated: chromatid and chromosome breaks, as well as chromatid and chromosome exchanges (i.e., rings, dicentric, and other polycentric chromosomes).

### 2.4. Statistical Analysis

The number of aberrations was counted. The frequency of aberrations was calculated and expressed as the mean and standard error. The X^2^ test and the Mann–Whitney U-test were used in order to compare experimental and control data. An analysis of dicentric distribution was performed with the free software CABAS V2.0, which was developed by Deperas et al. [18].

## 3. Results

### 3.1. Unstable Chromosome Analysis

Unstable chromosome aberrations were analyzed in peripheral blood lymphocytes in non-cancer patients and in the control group following CT scan examinations. The types and frequency of the detected chromosome aberrations are presented in Table 2.

Chromatid-type aberrations were found more frequently in the control group with a frequency of 0.75 ± 0.20%. Further, chromatid breaks were the main type of damage found among them 0.68 ± 0.20%. Chromosome-type aberrations were found with a frequency of 0.29 ± 0.08% and the ratio of the number of chromatid-type to chromosome-type aberrations was 3.05, which corresponds to the literature data on the prevalence of chromatid aberrations in control populations [19]. Dicentric chromosomes were detected in 27% of the persons in the control group. However, other polycentric or ring chromosomes were not detected within this group. The average dicentric chromosome frequency was 0.11 ± 0.04% (Table 2). The frequency of total aberrations measured in the control group, as well as chromatid and chromosome types, correspond to the literature data [19].

Chromosome-type aberrations had significantly higher frequency in the group of exposed patients. The ratio of the number of chromatid to chromosome aberrations was 0.71, which was significantly different from control group (X^2^ = 31.829 and *p* < 0.001).

Significant differences (*p* < 0.001) in the chromosome type breaks frequency between the exposed group and the control group were observed (0.38 ± 0.05% and 0.20 ± 0.08%, respectively). Among the chromosome breaks, small chromosome fragments called “minutes” were found. This type of aberration was often accompanied by dicentrics or was found separately (sometimes there were two or more per cell).

Dicentrics were detected in 80.1% (34) within the exposed patient group, which was statistically higher than that of the control group (*p* < 0.001). Further, rings and tricentric chromosomes were detected in 6 cases (14.3%). Cells with more than two dicentrics were frequently observed (35% of cases). Moreover, the frequency of dicentrics plus the rings estimated in exposed patients was significantly higher (0.53 ± 0.12%, *p* < 0.001) than that of the control group (Table 2).

Variations in the frequency of dicentric and ring chromosomes were identified: some patients demonstrated an increased number of dicentrics and rings after irradiation, while some patients showed the background dose level of damage. Fourteen persons demonstrated the control level of dicentrics (0–2 per 1000 cells), which corresponded to the data that is found for non-irradiated persons [13].

### 3.2. Dicentric Distribution in Patients after CT Examination

The distributions of dicentrics plus rings were estimated among the cells in accordance with the guidelines recommended by the International Atomic Energy Agency (IAEA) [13]. The results of distribution analysis are presented in Table 3. The dispersion index (the ratio variance to mean) and U-value were assessed. Sixteen patients indicated U-values above 1.96. This suggests an underdispersion of data and, correspondingly, non-Poisson distribution, which reflects heterogeneous exposure to radiation [13]. Overdispersion of data is in agreement with the partial-body exposure that occurs during a CT examination. Underdispersion (U-values below −1.96), which may be indicative of a problem in the data sampling, was not found. The analysis revealed that 18 patients indicated dispersion indices close to 1, as well as U-values of ±1.96, which indicates that the observed dicentric yields follow Poisson distribution and may reflect homogenous radiation exposure. Further, Poisson distribution was estimated in patients despite the fact that it was a partial-body radiation exposure.

## 4. Discussion

In the present study, a cytogenetic analysis of peripheral blood lymphocytes was performed in non-cancer patients after CT examination and non-cancer patients naïve to CT examination. The age distribution of the patients corresponded to the age of the most frequently CT-examined patients [20]. The sex ratio demonstrated a prevalence in the number of men within the exposed patient group (62.2%) and in the control group (77.2%) also. This, however, corresponded with the statistical data on the gender distribution of patients with the same types of CT examinations (e.g., abdomen, chest, and head) [21].

The analysis of the unstable chromosome aberrations was used to assess the genotoxic effects in patients after CT examination. The results demonstrated different patterns of genome damage. The prevalence of chromatid-type aberrations was found in the control group. Furthermore, this suggested prevalently mutagenic effects of the chemical factors. The results show that the control patients had a higher frequency of chromatid-type aberrations (0.75 ± 0.20%) than exposed patients (0.59 ± 0.07%). However, this difference was not statistically significant and corresponded to the background dose level [19]. An analysis of individual cytogenetic data revealed in the three control subjects an increased level of chromatid breaks that caused an increase in the mean value of chromatid aberrations for the control group. It may be suggested that these patients underwent some additional low-level chemical occupational exposure that they did not declare or experienced certain types of lifestyle exposure, as previously described [14,22]. Patients with an increased level of chromatid-type aberrations, however, were not found among irradiated persons.

When compared to controls, radiation–induced chromosome types of aberrations (such as in the case of dicentrics and rings) and chromosome breaks were detected in exposed patients, thereby demonstrating DNA damage that was induced after low dose medical irradiation [14,23]. The frequency of dicentrics and rings estimated in exposed patients was significantly higher than that of the control group. Individual variations in the frequency of dicentric and ring chromosomes, as well as distributions of dicentrics plus rings among cells were identified: some patients demonstrated an increased number of dicentrics and rings after irradiation, while some patients showed the background levels of damage; further, some patients demonstrated non-Poisson distributions of dicentrics, while others showed Poisson distribution. These variations may have several explanations and assumptions that should be considered. For example, a very small proportion of the blood lymphocytes were in the exposure field, such as in the case of CT scans of the brain that were diluted by non-irradiated cells, as well as certain damaged cells that were not found to have been analyzed. It is also impossible to exclude the fact that severely affected cells were eliminated or did not enter mitosis due to mitotic delay of the cell cycle as a result of DNA damage. The different types of distribution could be related to the history of medical exposure where: some patients could be irradiated once and others perform repeated examinations. Repeated examinations are, also, often performed in practice, as it was demonstrated by studies carried out in several countries [4,24,25]. Consequently, repeated CT examination leads to an increase in the cumulative effective dose of 100 mSv or greater, an increase in the number of chromosome aberrations, and the redistribution of irradiated lymphocytes among previously irradiated cells. The processes in regard to the induction of new aberrations, elimination of previous ones, and the delay of the mitotic cell cycle occur simultaneously, which can lead to a change in the patterns of dicentric distribution among cells. It is well known that dicentrics, no matter the fact that the type of radiation that induced them are mitotically unstable, are gradually eliminated from the turnover because they are unable to pass through repeated cell divisions. However, the phenomenon of a long time persistence of dicentrics was demonstrated within the different groups, as well as in accidentally exposed persons. Some of the patients were A-bomb survivors from Hiroshima and Nagasaki [26], persons exposed to the radiological accident in Goiania (Brazil) [27], Chernobyl cleanup workers [28], and Japanese fishermen who were exposed to fallout radiation from the nuclear explosion test site at Bikini Atoll [29]. Despite the difference in doses between victims of nuclear accidents and patients after diagnostics procedures, it may be speculated that some dicentrics after CT examination could reflect some previous exposure and, therefore, add a degree of variation to the results.

The investigation of the genotoxic effects of exposure to ionizing radiation during CT scans showed an elevated level of the g-H2AX, thereby suggesting that the CT examination dose was sufficient to induce DNA double-strand breaks in cells that correlated with the radiation dose [30,31,32,33,34]. There are only a few reports of cytogenetic effects after CT scanning. Cancer and non-cancer patients were cytogenetically investigated after single or repeated CT scans, before and after CT examination [35,36,37,38]. Variations in the frequency of dicentric and ring chromosomes were identified. Furthermore, some patients demonstrated an increased number of dicentrics after irradiation, but other patients showed lower levels of damage after a CT scan than before irradiation. These findings should allow researchers to conclude individual and complex mechanisms of induction, as well as the elimination of chromosome aberrations that could be connected with individual sensitivity, diseases, previous irradiation, or treatment [36,38]. When compared to these investigations, this study showed statistically increased levels compared to the control level of chromosome type aberrations (i.e., dicentrics, rings, and chromosome fragments) in group of patients after CT scanning. These patients did not suffer from oncological diseases and did not have chemotherapy or radiotherapy. Therefore, these aberrations could not be considered as the result of a combined effect of diseases, or radio- and chemotherapy. As such, this suggests there were genotoxic effects to CT examination.

A previously conventional unstable chromosome aberration analysis was performed on patients after CT scanning. Further, a significant (*p* < 0.0001) increase in chromosome aberration frequency was compared to that before exposure was found [37]. Only up to 300 cells were counted per persons. Furthermore, authors presented chromosome aberration yields similar to a sum of different types of chromosome aberrations (i.e., dicentrics, chromatid break, chromosome break, etc.) and did not specify the number of dicentrics. As such, there were not enough cytogenetic signs of overexposure that were found at this group. These results were obtained by the analysis of DNA damage before and after patient irradiation. Therefore, our work could be considered as the first case–control investigation of CT-exposed persons using conventional unstable chromosome analysis that demonstrated the genotoxic effects of low doses of ionizing radiation.

## 5. Conclusions

Through our results, the importance of lifelong cumulative exposure monitoring of ionizing radiation for diagnostical or therapeutical purposes is emphasized. Such records are important in the selection of diagnostic procedures, especially in oncology, due to the practice of repeated CT scans and pediatrics, which occur as a result of the possible long latency period of cancer risk. In the future, additional tests may be developed, by which radiosensitive persons may be recognized. Moreover, in this case, specific biomonitoring may be applied. It is crucial that clinicians are well informed regarding the level of genome damage caused by diagnostical procedures based on ionizing radiation so they can more actively participate in the optimization of patients’ exposures. A significant increase in the chromosome type of aberrations was detected in patients after CT scanning, suggesting that received doses represent possible health risks. Different types of dicentric distribution among the cells were found, thereby implicating complex mechanisms of chromosome aberration induction and elimination after exposure to low doses of medical irradiation. The results of this study: (a) confirm the occurrence of genome damage after CT scanning; (b) show the need for additional research; and (c) indicate that an improvement in CT facilities toward a decrease in dosage in order to reduce the cancer risk in patients, especially for those who require repeated CT scans, is required.

## Figures and Tables

**Table 1 life-12-01983-t001:** Baseline characteristics of patients.

N	Gender	Age	Part of Body Examined in CT	Effective Dose (mSv)	Scan with Contrast
1	Female	31	Abdomen	2.5	No
2	Male	32	Abdomen	3.1	No
3	Female	53	Abdomen	3.5	No
4	Female	33	Abdomen	3.7	No
5	Male	65	Abdomen	4.4	Yes
6	Male	45	Abdomen	4.7	No
7	Male	53	Abdomen	5.0	No
8	Male	49	Abdomen	8	Yes
9	Male	50	Abdomen	9.7	Yes
10	Male	44	Abdomen	9.9	Yes
11	Male	37	Abdomen	15	Yes
12	Female	42	Chest	1.5	No
13	Female	39	Chest	1.8	No
14	Male	71	Chest	2	No
15	Male	72	Chest	2	No
16	Male	70	Chest	2.2	No
17	Male	69	Chest	2.4	No
18	Male	54	Chest	2.6	No
19	Male	34	Chest	2.7	No
20	Female	51	Chest	2.8	No
21	Female	46	Chest	2.8	No
22	Female	72	Chest	2.8	No
23	Female	50	Chest	2.9	No
24	Male	54	Chest	3.0	No
25	Male	20	Chest	3.0	No
26	Male	40	Chest	3.0	No
27	Male	56	Chest	3.0	No
28	Male	45	Chest	4.0	Yes
29	Male	32	Chest	4.1	Yes
30	Male	48	Chest	4.1	Yes
31	Male	48	Chest	5.7	Yes
32	Male	42	Chest	5.8	Yes
33	Male	64	Chest	7.0	Yes
34	Male	64	Chest	7.0	Yes
35	Female	47	Chest	8.9	Yes
36	Female	74	Chest	9.2	Yes
37	Female	65	Chest	9.7	Yes
38	Female	55	Chest	9.8	Yes
39	Male	61	Chest	10.0	Yes
40	Female	52	Brain	2	No
41	Female	25	Brain	2.8	No
42	Male	46	Brain	3	No

**Table 2 life-12-01983-t002:** Unstable chromosome aberrations in peripheral blood lymphocytes of patients after CT examination and control group.

Frequency	Exposed Group, N = 42, M ± SE, %	Control Group, N = 22, M ± SE, %
**Chromatid-type aberrations**	0.59 ± 0.07	0.75 ± 0.20
Chromatid breaks	0.54 ± 0.07	0.68 ± 0.20
Chromatid exchanges	0.05 ± 0.01	0.07 ± 0.03
**Chromosome-type aberrations**	0.96 ± 0.14 **	0.29 ± 0.08
Chromosome breaks	0.38 ± 0.05 **	0.20 ± 0.08
Dicentrics + rings	0.53 ± 0.12 **	0.11 ± 0.04
**Total aberrations**	1.56 ± 0.14 *	1.06 ± 0.21

M: mean and SE: standard error. Statistically different from control group: * *p* < 0.05 and ** *p* < 0.001.

**Table 3 life-12-01983-t003:** Dicentric distribution in patients after CT examination.

N	Aberrations per Cell	Dispersion Index	U-Value	Poisson Distribution
1	0.002	2	31.83	No
2	0.003	3	57	No
3	0.004	2	22.89	No
4	0.005	2.2	29.9	No
5	0.006	2.2	28.3	No
6	0.006	4.3	83.1	No
7	0.006	4.3	83.1	No
8	0.008	2.6	32.1	No
9	0.007	1.32	7.62	No
10	0.01	4.26	76.9	No
11	0.01	3.76	67.4	No
12	0.034	4.22	72.8	No
13	0.016	1.19	3.45	No
14	0.017	5.16	82.15	No
15	0.044	6.58	62.44	No
16	0.006	3.33	57.94	No
17	0.0009	1	0	Yes
18	0.0009	1	0	Yes
19	0.0009	1	0	Yes
20	0.001	0.99	−0.031	Yes
21	0.001	0.99	−0.031	Yes
22	0.001	0.99	−0.031	Yes
23	0.002	0.99	−0.031	Yes
24	0.002	0.99	−0.03	Yes
25	0.002	0.99	−0.03	Yes
26	0.003	0.99	−0.05	Yes
27	0.003	0.99	−0.05	Yes
28	0.003	0.99	−0.055	Yes
29	0.003	0.999	−0.055	Yes
30	0.003	0.999	−0.055	Yes
31	0.004	0.998	−0.05	Yes
32	0.004	0.99	−0.093	Yes
33	0.005	0.99	−0.097	Yes
34	0.005	0.99	−0.097	Yes

## Data Availability

The data that support the findings of this study are not publicly available, due to patient confidentiality.

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
