# Peer review of "Cytogenetic Effects in Patients after Computed Tomography Examination"

_life, 2022, doi:10.3390/life12121983_

Round 1
Reviewer 1 Report (Previous Reviewer 2)
The authors report potential genome damage effects after CT examination in 42 non-cancer patients and 22 control subjects. Unstable chromosome aberrations analysis in peripheral blood samples shows a significant increase in abnormal chromosome types after CT imaging. The authors made substantial changes to the manuscript, and I recommended the manuscript for publication in Life after the authors addressed the following minor issues:
1. On page 5, line 183, the authors claim, "Prevalence of chromatid type aberrations was found in controls that suggested prevalently mutagenic effects of chemical factors." However, the difference is not statistically different. So how can the authors infer that the control group is subject to additional environmental effects?
2. On page 2, line 46, "radiation dose from 2 mounts to 10 years" should be "radiation dose from 2 months to 10 years".
Author Response
Response to Reviewer 1
Comment
- On page 5, line 183, the authors claim, "Prevalence of chromatid type aberrations was found in controls that suggested prevalently mutagenic effects of chemical factors." However, the difference is not statistically different. So how can the authors infer that the control group is subject to additional environmental effects?
Response
As general population is mostly exposed to chemical mutagens from source such as air pollution, food contaminants etc it is well known that in general population chromatid breaks are prevalent. Results are additionally explained and relevant references are added.
Comment
- On page 2, line 46, "radiation dose from 2 mounts to 10 years" should be "radiation dose from 2 months to 10 years".
Response
Typing mistake corrected.
Reviewer 2 Report (New Reviewer)
Neronova and Aleksanin focused on the cytogenetic effects of Computed Tomography (CT) examination in patients. They found CT scans may cause genome damage and may increase the cancer risk. This is an interesting study, but the results from this manuscript are not solid. The conclusion is vague.
The authors analyzed the chromosomal aberrations of peripheral blood from patients after CT scanning. They did those experiments after culturing cells from peripheral blood for a certain time and then detected the chromosome aberrations. During the culturing, the cells could repair their damaged chromosomes. The authors attributed the aberrations to merely CT scanning but did not consider the potential repair defects in the participants.
Author Response
Response to Reviewer 2
Comment
The authors analyzed the chromosomal aberrations of peripheral blood from patients after CT scanning. They did those experiments after culturing cells from peripheral blood for a certain time and then detected the chromosome aberrations. During the culturing, the cells could repair their damaged chromosomes. The authors attributed the aberrations to merely CT scanning but did not consider the potential repair defects in the participants.
Response
Time of cell culture for chromosome aberration assay is strictly defined and this method is official standardized method of biodosimetry in case of nuclear accident or overexposures recommended by International Atomic Energy Agency as cited in manuscript (Cytogenetic Dosimetry: Applications in Preparedness for and Response to Radiation Emergencies IAEA, Vienna (2011). DNA repair after irradiation starts in cell within minutes and this is known for decades and thus it was not mentioned in manuscript (Neumaier et al., 2011). As written in the manuscript “In exposed patients, blood was sampled within 3-14 days after CT“. Thus DNA rapir during cell culture was impossible. For purposes of DNA repair estimation specific methods are developed so called challenge assays in which DNA repair capacity is tested but this was not the aim of the current study. In order to eliminate possible residual repair proces patients were sampled not earlier than 3 days after exposure. This is explained in manuscript and relevant reference is added.
Reviewer 3 Report (New Reviewer)
The authors performed a study on chromosome examinations after CT-scans in non tumor patients compared to a control group.
Unfortunately, the study is very weak and the text should be improved.
There is neither a clear description about the detailed patient characteristics for each patient nor a result of the individual results for each patient.
To my knowledge it measure dicentrics at this very low level of only a few mSv. 1200 metaphases are far not enough.
Use of English language is not appropriate. Please let`s check by a native speaker.
Just some detailded comments:
Abstract line 18
“suggested compli-18 cated mechanisms of chromosome aberration induction”
What do the authors mean with this statement – please specify.
line 29 “It is estimated that 80,1% of the total radiation dose”
What is the scientific background of this assumption? Please cite. If this only holds true for Russia, please indicate.
Line 45: “background radiation dose from 2 mounts to 10 year” – again which background – which country – please specify
Line 49: “relative to similar one” – I don`t understand the phrase, rephrasing might be necessary
Material and methods:
2.1 Participants: did you rule out that the patient had any radiological procedures before. How about other risk factors for genotoxic stress?
2.2. CT scans.
Reported doses are very low and very hard to detect by chromosome aberration analysis.
Furthermore, the irradiation setup should be described more in detail. In particular the method how to assess the effective dose needs more attention. There is anyway a table missing to describe all individual findings including effective doses for every patient, kind of medical examand like the detailed information on the cytogenetic findings already shown..
2.3. Chromosome analysis: 1200 Methaphases are usually not sufficient to detect radiation induced chromosome aberrations after only a few mSv. Futhermore, this patient had a natural background radiation of more than 20-40 mSV (assuming 2 mSv/a)
Results:
Table 1: wrong use of punctuation
How were chromatid aberrations and even exchanges evaluated – with Giemsa stained methaphases? Was this Giemsa banding exame. I am realy very impressed, that must be a massive workload and is a clear sign, that the missing points adressed in this review should be added and the manuscript resubmitted.
4 fold increase due to 3-5 mSv in the rate of dicentrics is very hard to believe.
Table 2:
How do you explain the high yield of DIC of 0.034 to 0.044 in two patients? This should shift the whole statistical findings and I don`t understand how you could get this yield in otherwise healthy patient just with a standard CT exam.
Discussion:
191: “it may be suggested that these patients underwent some additional chemical occupational or life-style exposures” ? DIC are regared rather specific – what exactly is your assumption and how is the literature about it. Furthermore why were these points not ruled out and recorded in a “low dose study”?
198: “Breaks were most often detected in cells simultaneously with radiation markers that correspond to the ideas about the mechanisms of formation of radiation-induced chromosome aberrations” – I don`t understand this phrase. It should be explained in detail.
207: “for example very 207 small proportion of the blood lymphocytes have been in the exposure field”? I didn`t see any data of the authors to support this point. Please show in the results section. Anyway the patient characteristics, the field data, the effective dose, etc. are all missing.
215: “Consequently, repeated CT examination lead to an increase in the cumulative effective dose of 100 mSv or greater….” – with even the highest doses shown in this study (abd. With contrast) you need at least 7 examinations in a row to reach 100 mSv effective dose…. – What is the porpose of this statement? Diagnostic CT is not radiation therapy and not interventional diagnostics.
218-230: strange argumentation – not helpful at all. The incidents and accidents are very complex and to draw conclusions from high dose irradiations including neutron radiation, ingestion of radionuclides including alpha emitors to “small” doses of low LET irradiation should be avoided or handled with extreme care.
223: gH2AX and some other techniques are in my opinion much more adequate due to sensitivity to examine low doses of CT. Within these studies you can get the blood very fast after the irradiation and gH2AX is very sensitive.
250 “extremely significant“ – please avoid phrases like this. It`s a statistic test and the numbers are less informative. Significant is significant.
Author contribution:
What was the role of S.A.?
Did someone perform the experiments? Not mentioned.
Literature:
Please be homogenious and correct – see 17, 12, 11….
Author Response
Response to Reviewer 3
Comment
There is neither a clear description about the detailed patient characteristics for each patient nor a result of the individual results for each patient.
Response
Detailed characteristics for each patient are presented in Table 1. Individual results are given in Table 3.
Comment
To my knowledge it measure dicentrics at this very low level of only a few mSv. 1200 metaphases are far not enough.
Response
Number of cells for reliable results in biodosimetry is standardized for decades (IAEA, 2011, cited in manuscript). Thus as written in above mentioned publication:
“For lower doses, where the number of available cells is not the limiting factor, a dose estimate could be based on about 500 cells. This may require 2–3 person-days at a conventional microscope, although in an emergency several people can collaborate in scoring replicate slides. For a low or zero dicentric yield, the confidence limits resulting from 500 scored cells are usually sufficient. The decision to extend scoring beyond 500 to 1000 or more cells depends on whether there is evidence of a serious overexposure justifying an extended analysis, or if the continued employment of a radiation worker is in jeopardy.“
That means 1200 cells is almost 2.5 times more than recommended. Number of cells suggested by IAEA is not arbitral but based on statistical calculations.
Comment
Use of English language is not appropriate. Please let`s check by a native speaker.
Response
The manuscript will be additionally corrected for English by a native speaker. Unfortunately, to date we have not been able to edit the language by a native speaker. We will resolve this issue.
Comment
Abstract line 18
“suggested compli-18 cated mechanisms of chromosome aberration induction”
What do the authors mean with this statement – please specify.
Response
Word “complicated” is replaced with “complex” which is more suitable.
Comment
line 29 “It is estimated that 80,1% of the total radiation dose”
What is the scientific background of this assumption? Please cite. If this only holds true for Russia, please indicate.
Response
These data were obtained as a result of collecting and analyzing information on the levels of exposure of the Russian population to various sources of ionizing radiation during 2003-2018 years. Data collection is carried out within the framework of state radiation-hygienic passportization and using Joint state system of control and accounting of the individual doses of the Russian Federation citizens.
UNSCEAR report is added additionally to reference, which is already in manuscript. Russia has the same range of doses as the rest of the world due to geological diversity.
Comment
Line 45: “background radiation dose from 2 mounts to 10 year” – again which background – which country – please specify
Response
Reference: Lin EC. Radiation risk from medical imaging. Mayo Clin Proc. 2010, 85, 1142-1146; is added as the reference 4. Russia has the same range of natural background dose as the rest of the world due to geological diversity.
Comment
Line 49: “relative to similar one” – I don`t understand the phrase, rephrasing might be necessary
Response
It means that risk of cancer is higher in children. Sentence gives correct data and it is correct grammatically.
Comment
2.1 Participants: did you rule out that the patient had any radiological procedures before.
How about other risk factors for genotoxic stress?
Response
Patients were interviewed before the study. Criteria for inclusion and exclusion of patients in the study are indicated in paragraph 2.1. Patients who did not meet these criteria were excluded from the study. Patients denied contact with genotoxic factors. Information about this has been added to the relevant section.
Comment
2.2. CT scans.
Reported doses are very low and very hard to detect by chromosome aberration analysis.
Furthermore, the irradiation setup should be described more in detail. In particular the method how to assess the effective dose needs more attention. There is anyway a table missing to describe all individual findings including effective doses for every patient, kind of medical exam and like the detailed information on the cytogenetic findings already shown.
Response
Detailed information on irradiation setup has been added. Table 1 including effective doses for every patient in inserted into the text.
Comment
2.3. Chromosome analysis: 1200 Methaphases are usually not sufficient to detect radiation induced chromosome aberrations after only a few mSv. Futhermore, this patient had a natural background radiation of more than 20-40 mSV (assuming 2 mSv/a)
The number of requested cells for reliable results is previously described. We are not able to understand which patient had natural background radiation of 20-40 mSv and how it could be assumed as 2 mSv/a.
Chromosome aberration assay is sensitive enough to detect exposures of less than 10 mSv (Griciene, 2007; Sasaki et al., 2001) what is known for several decades.
Comment
Table 1: wrong use of punctuation
Response
Punctuation corrected.
Comment
4 fold increase due to 3-5 mSv in the rate of dicentrics is very hard to believe.
Response
Conclusion of 4 fold increase based on mean value can not be extracted as distribution and variations should be added. As it was already described low doses of 5 mSv may cause formation of dicentric chromosome as low as 5 mSv (Abe et al., 2015, page 6).
Comment
Table 2:
How do you explain the high yield of DIC of 0.034 to 0.044 in two patients? This should shift the whole statistical findings and I don`t understand how you could get this yield in otherwise healthy patient just with a standard CT exam.
Response
Interindividual differences and variations are described in Discussion line 240-250 with related references in which the same phenomenon is detected. As in each such study small number of patients was analysed, current study is of great value for investigation of origin of such dispersion of results for future better monitoring of patients.
Comment
Discussion:
191: “it may be suggested that these patients underwent some additional chemical occupational or life-style exposures” ? DIC are regared rather specific – what exactly is your assumption and how is the literature about it. Furthermore why were these points not ruled out and recorded in a “low dose study”?
Response
Although we are not sure what means word “regared” we suppose it is regarded. Dicentric is not specific for any genotoxic agent physical or chemical and presence of dicentric gives no information whether it is formed as a consequence of exposure to chemical agent or radiation. This is known for decades and literally thousands of papers and books are written on this topic.
In low dose studies environmental exposures can not be ruled out and pure fact that patients after CT show significant increase of chromosome aberration show that this diagnostics increase genome damage above background values.
Comment
198: “Breaks were most often detected in cells simultaneously with radiation markers that correspond to the ideas about the mechanisms of formation of radiation-induced chromosome aberrations” – I don`t understand this phrase. It should be explained in detail.
Response
Chapter is rewritten.
Comment
207: “for example very 207 small proportion of the blood lymphocytes have been in the exposure field”? I didn`t see any data of the authors to support this point. Please show in the results section. Anyway the patient characteristics, the field data, the effective dose, etc. are all missing.
Response
Table 1 is included which gives insight in patients characteristics. 207 line sentence is better explained.
Comment
215: “Consequently, repeated CT examination lead to an increase in the cumulative effective dose of 100 mSv or greater….” – with even the highest doses shown in this study (abd. With contrast) you need at least 7 examinations in a row to reach 100 mSv effective dose…. – What is the porpose of this statement? Diagnostic CT is not radiation therapy and not interventional diagnostics.
Response
CT in some diagnostic procedures is applied every three months using contrast. Thus in case of breast cancer application of some drugs demand such diagnostics (abdomen+pelvis) when metastasis are present. Patients that way accumulate in very short time 100 mSv. For more information please check for example algorithm for Ibrance.
Comment
218-230: strange argumentation – not helpful at all. The incidents and accidents are very complex and to draw conclusions from high dose irradiations including neutron radiation, ingestion of radionuclides including alpha emitors to “small” doses of low LET irradiation should be avoided or handled with extreme care.
Response
Chapter is rewritten.
Comment
223: gH2AX and some other techniques are in my opinion much more adequate due to sensitivity to examine low doses of CT. Within these studies you can get the blood very fast after the irradiation and gH2AX is very sensitive.
Response
gH2AX measures total level of DNA breaks without insight in chromosome rearrangements. It is method which can not be used in retrospective biodosimetry. The short time window during which samples are usable is the most severe limitation (please see for more details Raavi et al., 2021). Additionally, method is not standardized as chromosome aberration assay which causes unreliable interpretation of results as foci with irregular size or morphology remain significant problem for automated counting, and the presence of overlapping foci can also cause detection errors (please see Xiao et al., 2020).
Comment
250 “extremely significant“ – please avoid phrases like this. It`s a statistic test and the numbers are less informative. Significant is significant.
Response
Word “extremely” is deleted.
Comment
Author contribution:
What was the role of S.A.?
Did someone perform the experiments? Not mentioned.
Response
The corresponding explanations are included in the text:
E.N.: collection of material, perform the experiments, data analysis, writing—original draft.
S.A.: research concept and design, methodology, resources, project administration, data curation, writing—review and editing.
Comment
Literature:
Please be homogenious and correct – see 17, 12, 11….
Response
References are corrected in concordance with the requirements of the journal.
Round 2
Reviewer 2 Report (New Reviewer)
The authors have appropriately addressed my concerns.
This manuscript is a resubmission of an earlier submission. The following is a list of the peer review reports and author responses from that submission.
Round 1
Reviewer 1 Report
The authors evaluated the cytogenetic effects of a CT scan in non-cancer patients compared with patients who did not undergo CT scanning (controls).
The materials and methods section lacks clarifications.
- "Unstable chromosome aberrations were investigated": the reader understands that it is the frequency of chromosome aberrations that is studied, but this is not clearly stated,
- inclusion and exclusion criteria for patients are not given,
- the mode of recruitment? is it random?
- the design of the study is not clearly established: is it an experimental study in a laboratory? is it a clinical study in the care setting?
- justify the choice of this age range in discussion,
- discuss the choice of having more men than women
- how were the controls chosen? are they matched? on what criteria?
I find it unfortunate that the study did not follow a self-controlled design, in which exposed patients are compared to themselves when they were not exposed (in a BEFORE/ AFTER form), which would have involved taking blood samples before and after CT scan, but would have made it possible to avoid potential risk factors or other exposures/ different mutagenic factors effect.
In the results section:
- Why were the results not given by type of exposure/part of the body exposed? this could provide more knowledge about the most radio-sensitive parts of the body
- Table 1 lacks detail. What is the unit of the numbers given? recall to which test the p-value is related
- a comparative table of exposed vs. non-exposed patient characteritics is highly recommended,
- line 120, the numbers given in the text do not correspond to those in the table
- Mixing the results and the discussion in the same section is not relevant : it makes the section very long, and the results are not highlighted,
- the statistical analyses are relatively simple, whereas the available data would deserve more in-depth analyses, in particular it is really a pity not to have made a dose-response relation for each zone of the exposed body
- a major criticism is the small number of patients included, which translates into low statistical power
- there is little or no discussion of why unexposed patients have a higher frequency of chromatid-type aberrations than exposed patients (although this is non-significant).
Reviewer 2 Report
The manuscript describes cytogenetic effects in non-cancer patients after low-dose CT examinations. The authors investigate different types of chromosome aberrations, including dicentrics, ring chromosomes, and chromosome breaks, as biomarkers of genetic damage in a person exposed to medical ionizing radiation. Results show unstable chromosome aberrations in peripheral blood lymphocytes in patients after CT examination have very different trends in the exposed and control groups. Compared with methods with an in situ hybridization with the centromere and telomere peptide nucleic acid probes, the current approach is operationally simple and economical and provides a specific number of dicentrics. However, while there are growing concerns about the increased risk of cancer after diagnostic radiation, the authors need to discuss the background radiation from the cosmic, environment, et al. In addition, in the results section, some descriptions do not match the table, and the authors need to discuss more study details for low dose CT exposure. In particular, the authors need to address the following issues:
- What type of patients are involved in the study? And does the patient type contribute to chromosome aberrations of low-dose CT?
- Is the body weight or BMI a factor of chromosome aberrations upon radiation exposure?
- In the introduction section, the authors should discuss background exposures and give the audience a better understanding of the potential risk of CT scans when compared with daily life activities.
- "The average dicentric chromosome frequency was 0,11±0,04% (Table 1)." but it shows "dicentrics+rings" in table 1. Please clarify the number in table 1, 0,68±0,20 in table 1 verses 0,68±0,20% in the context.
- The author states that "cause an increased number of somatic mutations in human populations and, further investigation is needed for understanding biological and medical consequences of such type of radiation exposure." This statement does not support the hypothesis that any radiation may increase potential risk of cancers.
- Please provide a conclusion section in the manuscript. Please provide full names for IAEA, A-bomb, and BEIR VII. Please check the reference details to ensure consistency, such as ref. 15.
Round 2
Reviewer 1 Report
Thank you for the responses. The authors have made several necessary corrections. Some of the points made in the cover letter should have appeared in the body of the article, which is a bit of a loss. Nevertheless, a lot of work needs to be done on the English language, as the understanding of the article depends on it... I would suggest to hire an English speaker to ensure a good level of English.
Author Response
Dear reviewer,
Thank you very much for your questions and comments.
They allowed us to analyze the results of our research more thoroughly and make the necessary changes.
We plan to answer some of the questions that were raised in your review in our next study.
In accordance with your recommendation, we have made corrections to the English language and we hope the article has become more understandable.
Reviewer 2 Report
The authors appropriately addressed most comments. However, the responses to the comments are poorly organized. In addition, it seems some responses are copied from the website in various formats. Please address question 2.
Author Response
Dear reviewer, thank you very much for your questions, comments and suggestions.
Sorry for responses to the comments that are looks like poorly organized. We assume that, there were some technical problems while sending the response. u
Below we present answer to question 2:
Is the body weight or BMI a factor of chromosome aberrations upon radiation exposure?
Response:
There is discussion on patients' body size and body mass index as major determinants of irradiation for the patients in the literature. Taking into account dose dependence of chromosome aberrations it is possible to conclude that body weight and body mass index could be a risk factor for chromosome aberrations because of the higher dose irradiation. But in the literature, information on the effect of body weight on chromosome aberrations or other cytogenetic biomarkers as a result of diagnostic radiation exposure has not been found (for example, Habibi M, et al., “The Use of Genotoxicity Endpoints as Biomarkers of Low Dose Radiation Exposure in Interventional Cardiology”, Front Public Health. 2021 Jul 23;9:701878., or Shi L, et al., “Chromosomal Abnormalities in Human Lymphocytes after Computed Tomography Scan Procedure”, Radiat Res. 2018 Oct;190(4):424-432.).
We plan to test this assumption in the next our work.